# Studying the Effects of Varied Dosages and Grinding Times on the Mechanical Properties of Mortar

**Wenwen Zhang [1,2], Shujin Li [1,2,*], Luguang Song [2], Yanmin Sheng [2], Junwen Xiao [2] and Tianxiang Zhang [2]**

1    College of Environmental Science and Engineering, Changzhou University, Changzhou 213164, China
2    College of Civil Engineering and Architecture, Changzhou Institute of Technology, Changzhou 213032, China
*    Correspondence: lisj@czu.cn

**Abstract:** With the rapid development of construction and the construction industry, the demand for mortar as a building material is also increasing. With the development of economic society, glass products have been widely used, and glass manufacturing enterprises have produced hundreds of tons of glass fragments and slag. The main component of glass is silica, which has the potential to be used as an auxiliary cementing material. Therefore, waste glass is expected to be recycled in buildings to achieve sustainability. However, due to the chemical properties of the silica tetrahedral structure stabilized by the waste glass, its pozzolanic activity is potent and needs to be stimulated. Glass powders with different degrees of fineness were obtained by physical grinding of waste glass powder (WGP). The standard consistency water consumption, compressive strength, and flexural strength of waste glass powder cement mortar were studied. The grinding times of waste glass powder are 5 min, 10 min, and 15 min, respectively, and the dosages are 5%, 10%, 15%, and 20%, respectively. The experimental results show that the average particle sizes of the grinding times of 5 min, 10 min, and 15 min are 1670.0 μm, 243.0 μm, and 13.2 μm, respectively. The waste glass powder with a grinding time of 15 min has a specific surface area of 670 $m^2$/kg, which has high pozzolanic activity. The compressive and flexural strength of cement mortar specimens with 5% waste glass powder is the best, and the later strength is improved to a certain extent. The consistency of the cement mortar increased after adding waste glass powder. Compared with the 28 d compressive strength activity index of pure cement mortar specimens, the waste glass powder with 5–10% content reached more than 70%.

**Keywords:** waste glass powder; cement mortar; mechanical property; consistency

## 1. Introduction

Glass products are widely used in production and life because of their simple production process, light transmission, wear resistance, and low cost. With the development of the economy and society, the use of glass products is increasing yearly. Glass manufacturing enterprises produce hundreds of tons of cutting glass waste slurry and glass slag yearly. At present, the primary treatment method of waste glass slurry is disposal in a landfill. However, due to the stable chemical properties of glass, microbial degradation in the natural environment cannot be relied on, and landfill treatment is bound to cause serious environmental pollution problems. The mortar specimen is formed by mixing cement and sand as the primary raw material, in which cement plays an essential role in bonding as a cementitious material. However, it is also the raw material with the highest energy consumption, cost, and severe pollution in the mortar. The raw material of cement needs to be calcined at high temperatures, which consumes much energy, produces many greenhouse gases, and pollutes the environment. Cement is an essential raw material in building materials, and the market demand for cement forces much research on cement. Studies have found that the chemical composition of glass is similar to Portland cement,

and both contain relatively more silica [1–3], which can be used as a substitute for cementing material to reduce the consumption of Portland cement. Therefore, many studies have used waste glass powder as a cementing material for preparing mortar and concrete [1,4,5]. Resource utilization of waste glass powder is an effective way to solve the problem of glass processing, and glass powder is a volcanic ash material used to replace part of the cement, not only to reduce the amount of cement but also to save costs [6–8]. The use of waste glass for recycling in the construction industry [9–11] can reduce the burden of landfills and greatly benefit the conservation of resources.

Glass powder contains a large amount of active $SiO_2$. However, due to the chemical stability of glass, its pozzolanic activity is primarily potent, and appropriate methods need to be followed to stimulate it. At present, many scholars have carried out research on the related properties of glass powder applied to cement-based materials. Disfani et al. [12] showed that broken glass has excellent potential in cement products by studying the pozzolanic properties, hydration rate, and mechanical strength development of broken glass. Patel et al. [13] studied the essence and durability of glass powder with a particle size of 45 μm in the fresh, short-term, and long-term performance response of mixed mortar mix by analyzing glass powder with 0–20% substitution, and the results showed that glass powder as a substitute for cement is feasible. Ali et al. [14] studied the effect of waste glass powder on the strength of cement mortar by machine learning (ML) and Shapley additive explanations (SHAP) methods. The results showed that waste glass powder and cement positively affect the strength of cement mortar. Some scholars [15–17] have also shown that waste glass powder has the potential to be used as an auxiliary cementing material, and waste glass powder exhibits better ASR resistance. Oumaima et al. [18] showed that adding waste glass powder could significantly affect the thermal physical properties of ordinary cement mortar.

Many studies have demonstrated [19–26] that waste glass powder of appropriate particle size produced after further grinding broken glass can be used as a substitute for Portland cement. The pozzolanic activity of glass powder increases with the decrease in the particle size of waste glass powder, and the strength of cement mortar increases with the decrease in the glass powder particle size. Zhao et al. [27] also showed that glass powder with small particle sizes contributed to the development of the compressive and flexural strength of mortar specimens. Gholipour et al. [28] found that the pozzolanic activity of glass particles increased when the glass particles were ground to a particle size below 300 μm. The pozzolanic activity of waste glass powder ground to less than 100 μm is better than that of fly ash (FA). Zhu et al. [29] showed that the strength of alkali slag concrete increased with the decrease in glass powder particle size and glass powder content. Khatib et al. [30] found that when the amount of glass powder replacing cement was 10%, the compressive strength reached its maximum; when above 10%, the intensity decreased. The study by Rao [31] has shown that the strength of mortar specimens mixed with glass powder increases with age, and the strength of mortar specimens is optimal when the glass powder content is 10%. Nasir et al. [32] adopted waste glass powder to replace cement and fine aggregate through experiments and the machine-learning (ML) method. The results showed that flexural strength is the best when 10% and 15% waste glass powder replaces cement and fine aggregate. Nihat et al. [33] studied the application of pumice powder (PP) and glass beads (GM) in cement mortar by the slurry replacement method (PRM). The results showed that replacing 20% cement with glass powder did not affect its compressive strength or improve its physical properties.

The different degrees of fineness of glass powder were obtained by physically grinding waste glass powder. The grinding times of waste glass powder were 5 min, 10 min, and 15 min, respectively, and the dosages were 5%, 10%, 15%, and 20%, respectively. The standard consistency water consumption, compressive strength, and flexural strength of waste glass powder cement mortar were studied. The influence of different particle sizes and waste glass powder dosages on cement mortar's mechanical properties was evaluated by macroscopic test and microscopic analysis.

## 2. Raw Materials and Test Methods

### 2.1. Test Raw Materials

The raw materials of this study include cement (P ● O 42.5 ordinary Portland cement), sand (ISO standard sand, Xiamen Esio Standard Sand Co., LTD), waste glass powder (WGP) from a lamp processing enterprise in Changzhou City, Jiangsu Province, China, and water (tap water). The chemical composition of cement and waste glass powder is shown in Table 1.

**Table 1.** Chemical composition of cement and glass powder.

| Material | $Al_2O_3$ | $SiO_2$ | CaO | $Na_2O$ | $Fe_2O_3$ | MgO | $SO_3$ | LOI |
|---|---|---|---|---|---|---|---|---|
| Cement | 4.45 | 21.23 | 63.61 | 0.51 | 3.17 | 2.31 | 2.27 | 2.45 |
| WGP | 2.74 | 73.32 | 10.26 | 9.94 | 1.09 | 1.38 | 0.12 | 1.15 |

### 2.2. Grinding Treatment of WGP

The WGP was pre-cleaned to remove impurities. WGP was placed into an oven to dry at 80 °C for 4 h and then a cement ball mill was used as shown in Figure 1, where the WGP was ground for periods of 5 min, 10 min, and 15 min. The inner liner of the ball mill is a cylinder with an inner diameter of 500 mm and a length of 500 mm. The speed of the ball mill is 0.8 r/s, and the mass ratio of the steel ball to the WGP during grinding is 6:1. Figure 2 shows the appearance of WGP after ball milling.

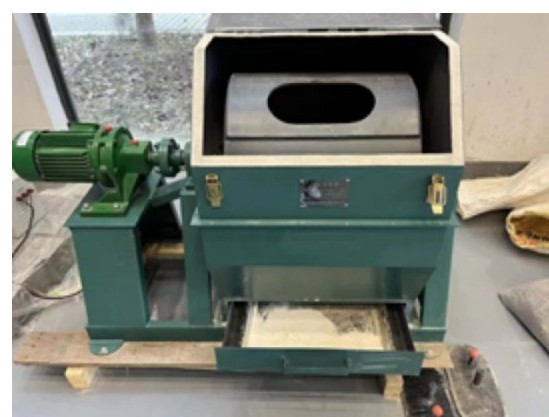
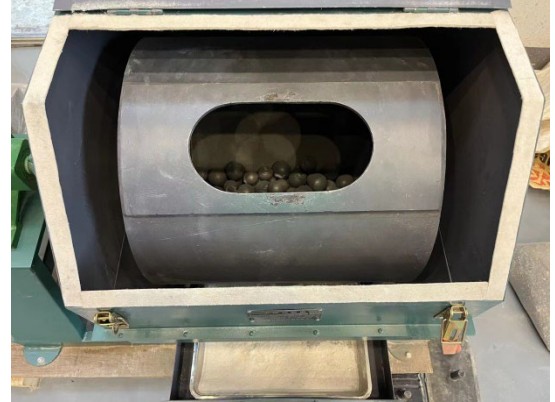

**Figure 1.** Cement ball mill.

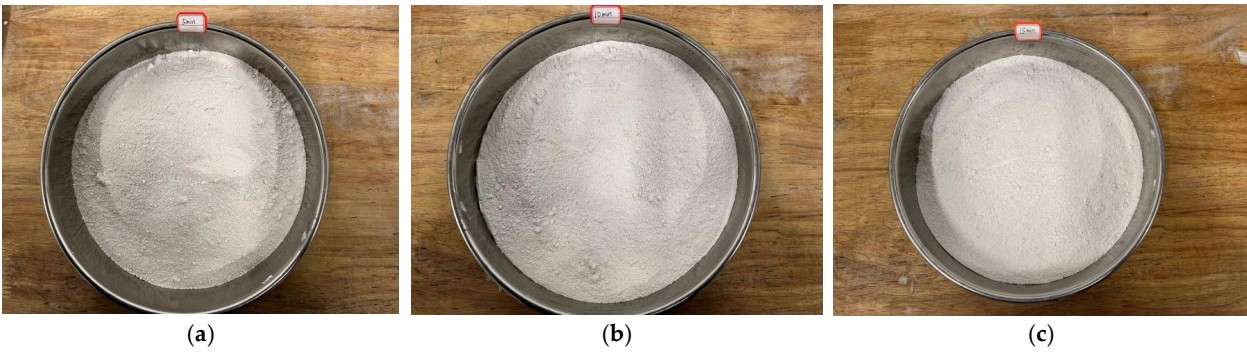

**Figure 2.** Appearance of waste glass powder. (**a**) 5 min. (**b**) 10 min. (**c**) 15 min.

### 2.3. Mix the Proportion of Cement Mortar

The mix proportion of test base mortar (PC) was cement, standard sand, and water = 450 g, 1350 g, and 225 g, respectively. The WGP was used to replace the cement to prepare the waste glass powder cement mortar specimen. The content of WGP was

5%, 10%, 15%, and 20%, respectively. The water consumption of each ratio was adjusted to achieve the same consistency as the benchmark ratio. The proportion of WGP cement mortar is shown in Table 2. F5, F10, and F15 represent the grinding times of adding WGP as 5 min, 10 min, and 15 min, respectively. C5, C10, C15, and C20 represent WGP dosages of 5%, 10%, 15%, and 20%, respectively.

**Table 2.** Mixture proportions of waste glass powder cement mortar.

| Group | Standard Sand (g) | Cement (g) | WGP (g) | Water (g) |
|---|---|---|---|---|
| PC | 1350 | 450.00 | 0.0 | 225.0 |
| F5C5 | 1350 | 427.50 | 22.5 | 231.2 |
| F5C10 | 1350 | 405.00 | 45.0 | 237.5 |
| F5C15 | 1350 | 382.50 | 67.5 | 243.8 |
| F5C20 | 1350 | 360.00 | 90.0 | 250.0 |
| F10C5 | 1350 | 427.50 | 22.5 | 231.2 |
| F10C10 | 1350 | 405.00 | 45.0 | 237.5 |
| F10C15 | 1350 | 382.50 | 67.5 | 243.8 |
| F10C20 | 1350 | 360.00 | 90.0 | 250.0 |
| F15C5 | 1350 | 427.50 | 22.5 | 231.2 |
| F15C10 | 1350 | 405.00 | 45.0 | 237.5 |
| F15C15 | 1350 | 382.50 | 67.5 | 243.8 |
| F15C20 | 1350 | 360.00 | 90.0 | 250.0 |

The forming, curing, and strong determination of cement mortar specimens were carried out following GB/T 17671-2021. The 40 mm × 40 mm × 160 mm prismatic specimens were prepared, cured for 24 h, and then removed. The specimens were then cured for 3 d, 7 d, and 28 d, respectively, in a water curing tank at 20 ± 1 °C. The compressive strength and flexural strength of the specimens were tested according to the standard methods.

## 3. Test Results and Discussion

### 3.1. Effect of Grinding Fineness on Mortar Consistency

The particle size distribution of WGP after grinding was characterized and tested by a Malvern laser particle size analyzer (Mastersizer 3000E). The particle size distribution curves of the WGP under different grinding times are shown in Figure 3. It can be seen that there are still many large particles in the waste glass powder after grinding for 5 min to form the double peaks of the particle size grading curve. Increasing the grinding time to 10 min, the volume density of the double peaks gradually decreases. The double peak of the particle size grading curve shifted toward smaller particle sizes, and the particle size distribution of waste glass powder formed a single peak distribution after 15 min of grinding. The particle sizes of the waste glass powder after grinding for 5 min, 10 min, and 15 min were 91.2–2680 µm, 4.87–889 µm, and 4.01–25.6 µm, respectively.

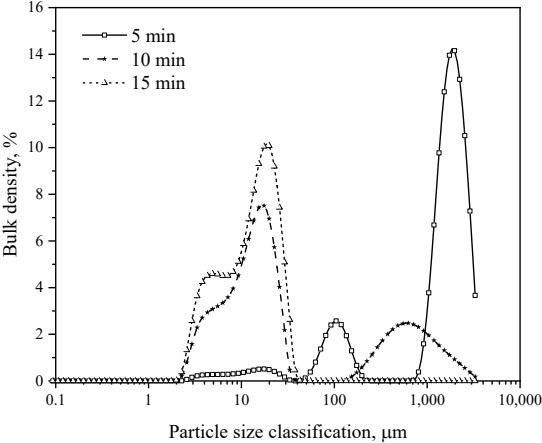

**Figure 3.** Particle size distribution of waste glass powder at different grinding times.

Table 3 shows the specific surface area and average particle size of waste glass powder after grinding for 5 min, 10 min, and 15 min. It can be seen that the specific surface area

of waste glass powder increased significantly after 15 min of grinding, and the average particle size reached 13.2 μm. This shows that the longer the grinding time, the better the grading of the waste glass powder. The increase in fineness leads to an increase in specific surface area, which helps waste glass powder participate in the reaction and improve the strength of waste glass powder cement mortar.

In the experiment, 500 g of P ● O42.5 ordinary Portland cement and an appropriate amount of water were used to measure the consistency value (sinking amount) of 32 mm. The water consumption of the corresponding standard consistency was 142.5 g, the initial setting time was 180 min, and the final setting time was 270 min. On the premise of constant water consumption, waste glass powder with different grinding times and dosages was used to replace part of the cement to measure the consistency value of each group.

**Table 3.** Specific surface area and average particle size of waste glass powder at different grinding times.

| Grinding Time (min) | Specific Surface Area (m²/kg) | The Average Particle Size (μm) |
|---|---|---|
| 5 | 49.17 | 1670 |
| 10 | 478.7 | 243 |
| 15 | 672.7 | 13.2 |

The influence of the consistency value of the cement mortar specimen with the grinding time of the waste glass powder at each dosage is shown in Figure 4. As can be seen from Figure 4, the consistency of cement mortar specimens, on the whole, decreases with the increase in grinding time. At 5%, with the increase in grinding time, the consistency of cement mortar remained unchanged and then decreased. Compared with the reference group, F5, F10, and F15 decreased by 26.6%, 26.6%, and 45.3%, respectively. At 10%, with the increase in grinding time, the consistency value of the cement mortar specimen is improved. At F5, the consistency value was the lowest, which decreased by 78.1% compared with the benchmark group. At 15%, the consistency value of the F15 cement mortar specimen decreased the most compared with the reference group, which was 98.4%. At 20%, F5, F10, and F15 decreased by 84.4%, 84.4%, and 78.1%, respectively, compared with the baseline group. It can be seen that the dosage has a great influence on the consistency of the cement mortar. This is because the specific surface area of waste glass powder is larger, and the greater the mixing amount, the greater the water demand. However, when the total water consumption is constant, the fluidity decreases and the consistency value decreases.

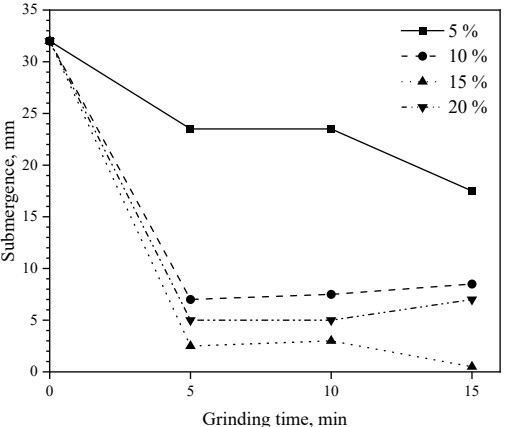

**Figure 4.** Effect of grinding time and dosage of waste glass powder on the consistency of cement mortar.

### 3.2. Effect of Grinding Time of Waste Glass Powder on Compressive and Flexural Strength of Mortar

Figure 5 shows the influence of different grinding times on the compressive strength of mortar specimens when the content of waste glass powder is 5%. Compared with the PC group, the compressive strength of F5, F10, and F15 decreased by 23.2%, 21.5%, and 11.3% at the age of 3 d, respectively. At 7 days, the compressive strength of F5, F10, and F15 decreased by 24.3%, 26.4%, and 12.8%, respectively, compared with that of the PC group. At 28 d, the compressive strength of F5, F10, and F15 decreased by 20.1%, 8.7%, and 11.4%, respectively, compared with that of the PC group.

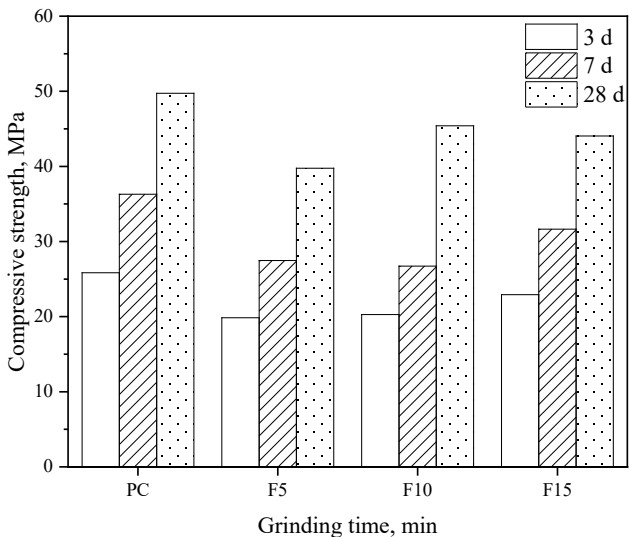

**Figure 5.** Effect of different grinding times on compressive strength of mortar—C5 group.

With the increase in grinding time, the fineness of the waste glass powder decreases, and the strength of the waste glass powder mortar specimens increases gradually. Compared with 3 d compressive strength, the strength of the PC group increased by 40.4% and 92.4% at 7 d and 28 d, respectively. The strength of the F5 group increased by 38.4% and 100.3% at 7 d and 28 d, respectively. The strength of the F10 group increased by 31.8% and 123.9% at 7 d and 28 d, respectively. The strength of the F15 group increased by 38.1% and 92.2% at 7 d and 28 d, respectively. It can be found that the strength development of different grinding times in the early stage is lower than that of the PC group, and in the later stage, except the F15 group, which is slightly lower than that of the PC group, the rest are significantly higher than that of the PC group. It can be seen that the compressive strength of each group increases to different degrees with the increase in age, and the improvement of compressive strength in the late stage is significantly higher than that in the early stage, which is mainly because the secondary hydration of waste glass powder needs to be carried out under alkaline conditions, and the time lags behind the hydration of cement.

Figure 6 shows the effects of different grinding times and reference groups, 5% waste glass powder content, and different ages on flexural strength. Compared with the reference group, the influence of waste glass powder cement mortar specimens on flexural strength at each grinding time is similar to that of compressive strength. At the same age, the flexural strength of the waste glass powder increases continuously with the increase in grinding time.

The change in flexural strength of mixed mortar mixed with waste glass powder was significantly higher than that of the reference group. The longer the curing time, the better the development of strength. In the PC group, the flexural strength of 7 d and 28 d increased by 14.8% and 41.7%, respectively, compared with 3 d. However, in the F5 group, the flexural strength of 7 d and 28 d was increased by 23.0% and 67.0%, respectively, compared with that of 3 d. In the F10 group, the flexural strength of 7 d and 28 d was increased by 22.5%

and 56.5%, respectively, compared with that of 3 d. In the F15 group, the flexural strength of 7 d and 28 d was improved by 35.5% and 49.9%, respectively, compared with that of 3 d.

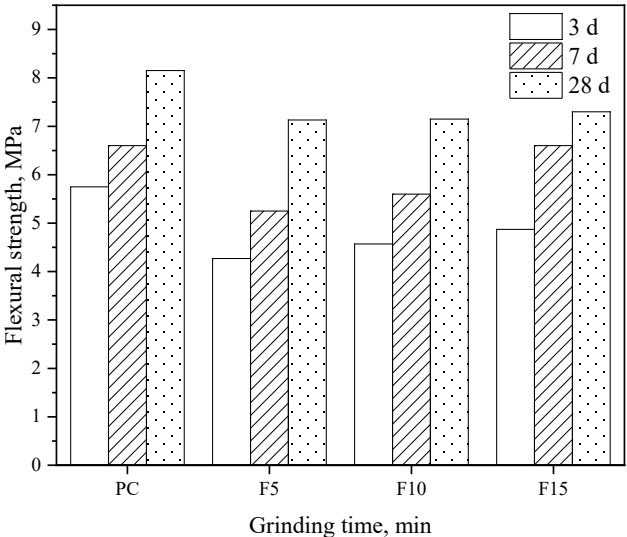

**Figure 6.** Effect of different grinding times on flexural strength of mortar—C5 group.

Figure 7 shows the influence of different grinding times on the compressive strength of mortar specimens when the content of waste glass powder is 10%. It can be seen that, at the same age, the compressive strength of waste glass powder mortar specimens with different grinding times has decreased by varying degrees compared with the reference group. With the increase in grinding times, the compressive strength of waste glass powder mortar specimens tends to change smoothly at various ages. Compared with 3 d compressive strength, the strength of the PC group increased by 40.4% and 92.4% at 7 d and 28 d, respectively. The strength of the F5 group increased by 58.7% and 130.6% at 7 d and 28 d, respectively. The strength of the F10 group increased by 40.2% and 126.8% at 7 d and 28 d, respectively. The strength of the F15 group increased by 41.2% and 96.5% at 7 d and 28 d, respectively. Except for the F10 group being slightly lower than the PC group in the early stage, the compressive strength of the different grinding times at various ages was generally higher than the PC group. This indicates that secondary hydration is developed to a certain extent with increased waste glass powder content.

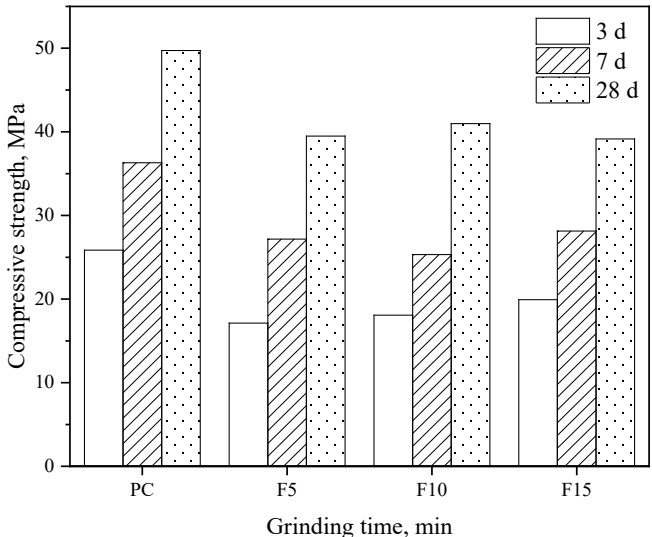

**Figure 7.** Effect of different grinding times on compressive strength of mortar—C10 group.

Figure 8 shows the influence of 10% waste glass powder content at different grinding times and reference groups on the flexure strength at different ages. Compared with the reference group, the effect of waste glass powder cement mortar specimens on flexural strength and compressive strength at each grinding time is similar. At the same age, the flexural strength of waste glass powder tends to stabilize with the increase in grinding times.

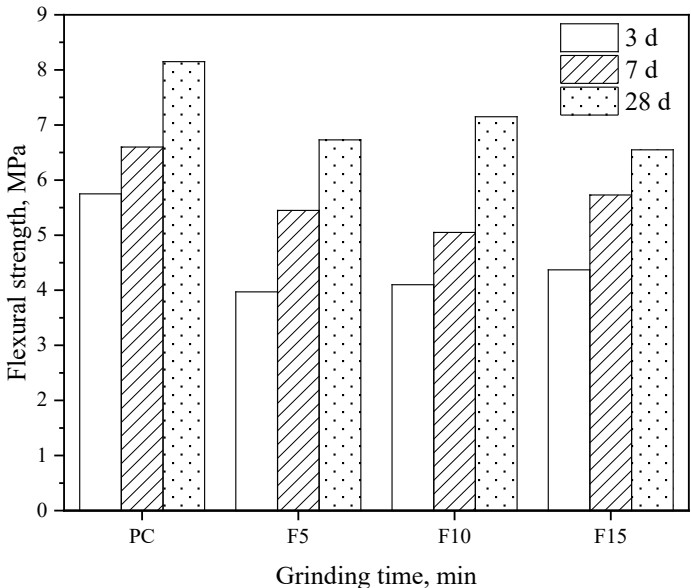

**Figure 8.** Effect of different grinding times on flexural strength of mortar—C10 group.

The flexural strength of mixed mortar mixed with waste glass powder changed significantly compared with the reference group. The longer the curing time, the better the development of strength. In the PC group, the flexural strength of 7 d and 28 d increased by 14.8% and 41.7% compared with 3 d, respectively. However, in the F5 group, the flexural strength of 7 d and 28 d increased by 37.3% and 69.5% compared with 3 d, respectively. In the F10 group, the flexural strength of 7 d and 28 d increased by 23.2% and 74.4% compared with 3 d, respectively. In the F15 group, the flexural strength of 7 d and 28 d increased by 31.1% and 49.9% compared with 3 d, respectively.

Figure 9 shows the influence of different grinding times on the compressive strength of mortar specimens when the content of waste glass powder is 15%. It can be seen that the compressive strength of waste glass powder mortar specimens with different grinding times is significantly lower than the reference group at various ages. At the age of 3 d, the compressive strength of F5, F10, and F15 decreased by 39.9%, 40.6%, and 36.2% compared with the PC group. At 7 d, the compressive strength of F5, F10, and F15 decreased by 38.1%, 35.8%, and 34.6% compared with the PC group. At 28 d, the compressive strength of F5, F10, and F15 decreased by 26.4%, 33.3%, and 30.4% compared with the PC group. It can be seen that under the 15% waste glass powder content, the compressive strength of mortar specimens under different grinding times is less than 70% of the PC group's strength at each age. This shows that excessive waste glass powder will affect the further reaction with calcium hydroxide, decreasing the strength.

Figure 10 shows the influence of different grinding times on the flexural strength of mortar specimens when the content of waste glass powder is 15%. Compared with the reference group, the influence of waste glass powder cement mortar specimens on flexural strength and compressive strength at various grinding times is similar. The flexural strength of waste glass powder mixed at 15% at different grinding times is generally lower than the reference group.

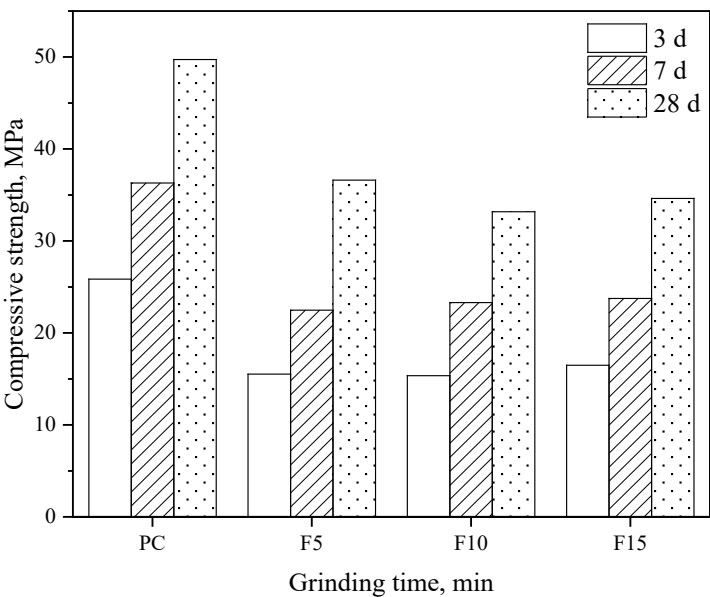

**Figure 9.** Effect of different grinding times on compressive strength of mortar—C15 group.

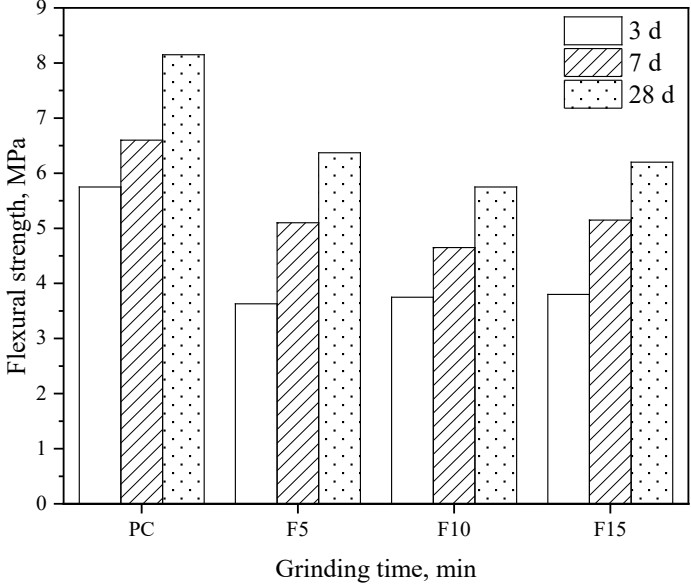

**Figure 10.** Effect of different grinding times on flexural strength of mortar—C15 group.

Figure 11 shows the influence of different grinding times on the compressive strength of mortar specimens when the content of waste glass powder is 20%. At the same age, the compressive strength of mortar specimens with different grinding times of waste glass powder at 20% content is lower than that of the reference group. At the age of 3 d, the compressive strength of F5, F10, and F15 decreased by 50.7%, 51.8%, and 38.0% compared with the PC group. At 7 d, the compressive strength of F5, F10, and F15 decreased by 45.0%, 47.7%, and 42.6% compared with the PC group. At 28 d, the compressive strength of F5, F10, and F15 decreased by 34.1%, 38.6%, and 35.9% compared with the PC group. Under the dosage of 20% waste glass powder, the compressive strength of mortar specimens under different grinding times is less than 70% compared with the PC group at each age, which further indicates that excessive waste glass powder content will affect the development of strength.

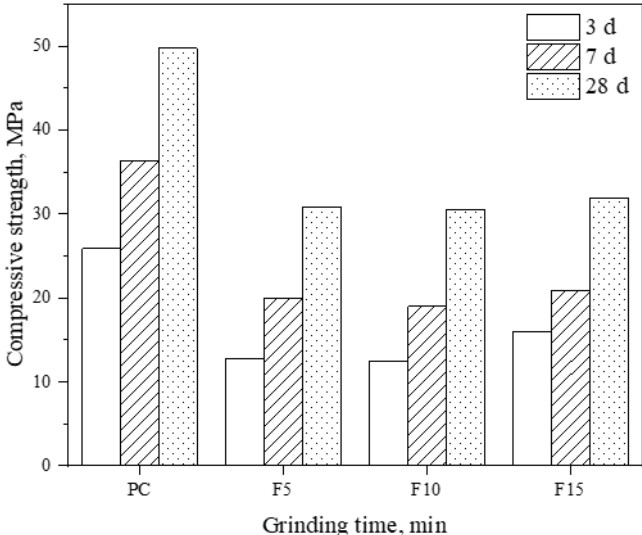

**Figure 11.** Effect of different grinding times on compressive strength of mortar—C20 group.

From the above, it can be observed that the increase in the content of waste glass powder has a great influence on the compressive and flexural strength of the cement mortar. In this test, the C5 group showed the smallest reduction in compressive and flexural strength compared to the PC group. However, the increasing ratio of compressive and flexural strength in group C10 and group C15 was significantly better than that in the PC group, and the C10 group played a pivotal role in the development of strength.

Figure 12 shows the influence of different grinding times on the flexural strength of mortar specimens when the content of waste glass powder is 20%. Compared with the reference group, the effect of waste glass powder cement mortar specimens on flexural strength and compressive strength at different grinding times is similar. The flexural strength of 20% waste glass powder at different grinding times is significantly lower than the reference group.

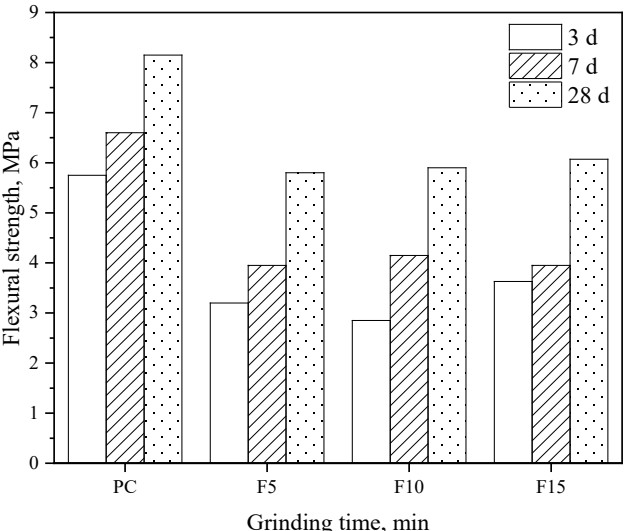

**Figure 12.** Effect of different grinding times on flexural strength of mortar—C20 group.

Through the above research, it can be found that the grinding time of waste glass powder has a positive effect on the compressive and flexural strength of cement mortar. The longer the grinding time, the smaller the average particle size of the waste glass powder and the better the flexural strength of the specimen. Sun et al. [34] also found that



mechanical grinding of waste glass powder from 300 μm to 75 μm effectively enhanced the compressive strength and reduced the expansion caused by the alkali–silica reaction. As far as this test is concerned, the waste glass powder cement mortar specimen of the F15 group has the lowest overall loss of compressive flexural strength. The C5 group lost the slightest compressive and flexural strength at each age. The C10 group had the best compressive and flexural strength development at the later stage. The C15 and C20 groups had no significant advantage in developing cement mortar's compressive and flexural strength. The increase in grinding time brings more waste glass powder to participate in the hydration reaction. The waste glass powder particles that did not participate in the hydration reaction can fill the gap of hydration products and form a more compact microstructure, thus contributing to the strong development of waste glass powder cement mortar specimens.

### 3.3. Effect of Waste Glass Powder Content on the Compressive and Flexural Strength of Mortar

Figure 13 shows the influence of different dosages on the compressive strength of mortar 3 d, 7 d, and 28 d under the premise of the F15 group. With the increase in waste glass powder content, the strength of the mortar specimen decreases gradually. At the same age, the compressive strength of mixed mortar with different dosages is lower than that of the reference group. Under 3 d, the compressive strength of C5, C10, C15, and C20 decreased by 11.3%, 22.9%, 36.2%, and 38.0% compared with the PC group, respectively. At 7 d, the compressive strength of C5, C10, C15, and C20 decreased by 12.8%, 22.5%, 34.6%, and 42.6%, respectively, compared with the PC group. At 28 d, the compressive strength of C5, C10, C15, and C20 decreased by 11.4%, 21.3%, 30.4%, and 35.9%, respectively, compared with the PC group. The changing trend of compressive strength of cement mortar specimens mixed with waste glass powder is generally better than that of the PC group, and the change of compressive strength is more significant with the increase in age. Compared with 3 d compressive strength, the strength of the PC group increased by 40.4% and 92.4% at 7 d and 28 d, respectively. The strength of the C5 group increased by 38.1% and 92.2% at 7 d and 28 d, respectively. The strength of the C10 group increased by 41.2% and 96.5% at 7 d and 28 d, respectively. The strength of the C15 group increased by 44.1% and 110.1% at 7 d and 28 d, respectively. The strength of the C20 group increased by 30.1% and 98.9% at 7 d and 28 d, respectively.

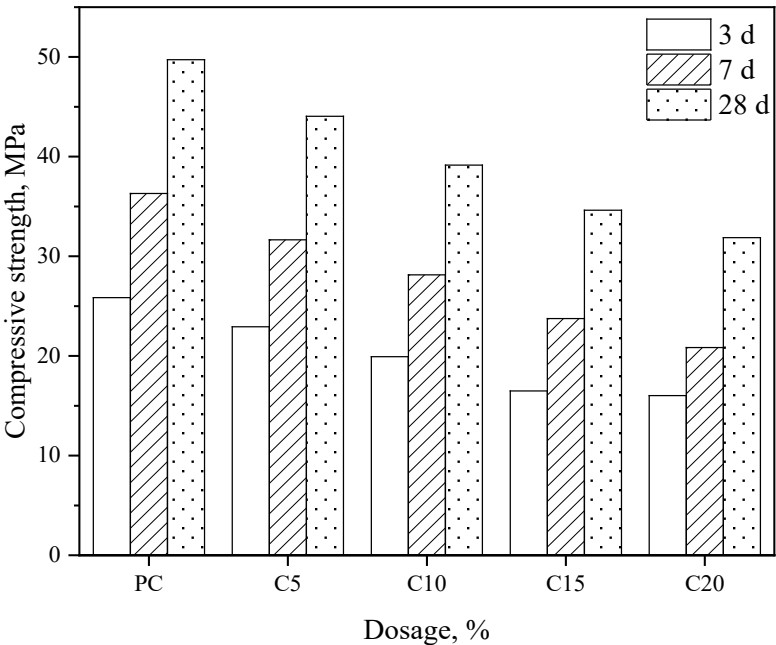

**Figure 13.** Effect of different dosages on compressive strength of mortar—F15 group.

Under the conditions of the F15 group, the influence of different dosages and reference groups at different ages on flexural strength is shown in Figure 14. With the increase in dosage, the flexural strength of waste glass powder cement mortar specimens showed a downward trend. With the increase in age, flexural strength shows an upward trend, similar to compressive strength.

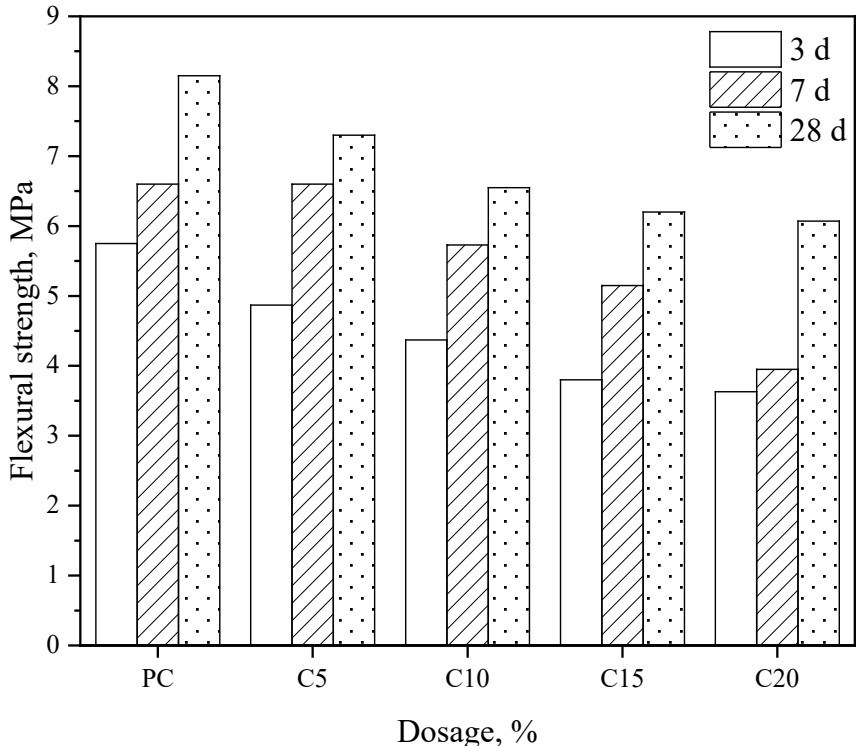

**Figure 14.** Effect of different dosages on flexural strength of mortar—F15 group.

With the increase in the content of waste glass powder, the alkali content increases. When the dosage is small, the alkali in waste glass powder reacts with the calcium hydroxide produced by cement hydration to form a certain C-S-H gel, which contributes to the strength of the cement mortar. However, too much waste glass powder will hinder the further reaction with calcium hydroxide, resulting in a decrease in strength.

Figure 15 shows the change of flexural strength with age under different mixing amounts at different grinding times. As can be seen from Figure 15, with the increase in the content, the flexural strength of each grinding time and each age generally decreased. However, with the increase in age, compressive strength is generally improved. However, there is a slight increase under some grinding times, dosages, and ages. As shown in Figure 15a, at 7 d, the flexural strength first increased and then decreased with the increase in dosage, and a turning point occurred at a 10% dosage. As shown in Figure 15b, at 28 d, the flexural strength first remained unchanged and then greatly decreased and then increased with the increase in dosage. As shown in Figure 15c, the flexural strength decreases with the increase in dosage at each age.

From the above, it can be observed that the increase in the content of waste glass powder has a great influence on the compressive and flexural strength of the cement mortar. In this test, the C5 group showed the smallest reduction in compressive and flexural strength compared to the PC group. However, the increasing ratio of compressive and flexural strength in the C10 group and C15 group was significantly better than that in the PC group, and the C10 group played a pivotal role in the development of strength.

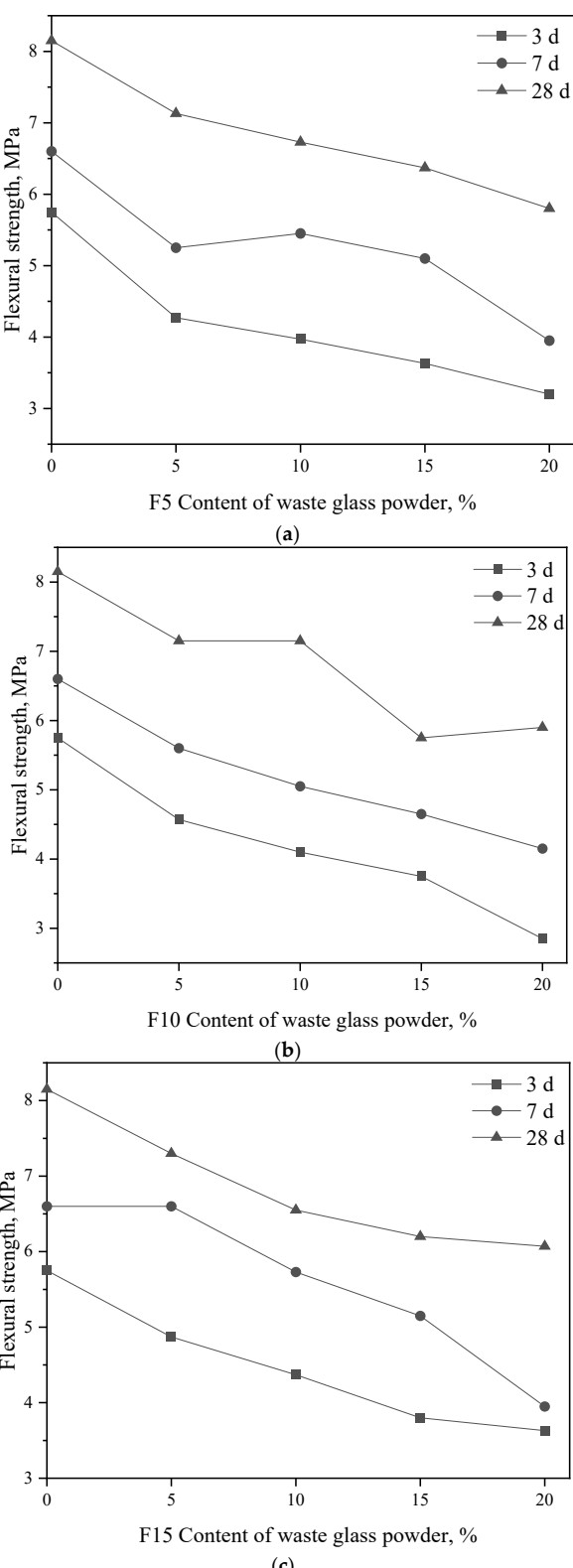

**Figure 15.** Effect of different grinding times and dosages on flexural strength with age. (**a**) F5 group. (**b**) F10 group. (**c**) F15 group.

### 3.4. Effect of Waste Glass Powder on the Collapse–Pressure Ratio of Mortar

The fold compression ratio is the ratio of compressive strength and flexural strength, and the size of the fold compression ratio reflects the merits of its crack resistance [35,36]. Figure 16 shows the change in the collapse–compression ratio of 3 d cement mortar spec-

imens with the amount of waste glass content at different grinding times. It can be seen that the collapsible ratio of waste glass powder cement mortar specimens in group F5 first increases and then decreases with the increase in mixing amount. The folding ratio of F5C5 was the highest, which increased by 3.3% compared with the reference group. The folding pressures of F10C5, F10C10, F10C15, and F10C20 were reduced by 1.3%, 2.0%, 9.1%, and 4.4%, respectively, compared with the reference group. The folding pressure ratio of F15 waste glass powder cement mortar specimens increased first and then decreased and increased again with the increase in dosage. The folding pressure of F15C5 and F15C10 increased by 4.7% and 1.3%, respectively, compared with the reference group. The folding pressure of F15C15 and F15C20 decreased by 3.6% and 2.0%, respectively, compared with the reference group.

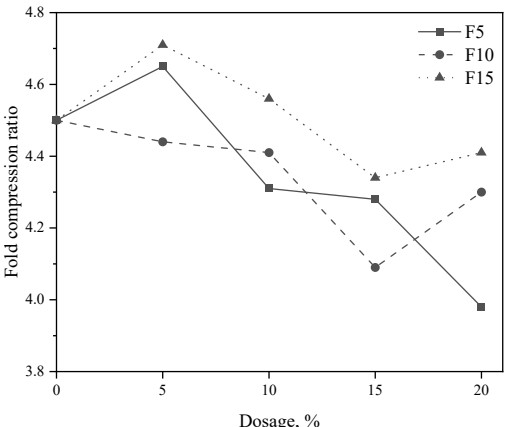

**Figure 16.** The fold compression ratio of waste glass powder cement mortar specimens—3 d.

Figure 17 shows the change in the compressive ratio of the 28 d waste glass powder cement mortar specimen with the waste glass content at different grinding times. The folding pressures of F5C5, F5C10, F5C15, and F5C20 were 8.5%, 3.8%, 5.7%, and 13.0% lower than those of the reference group, respectively. The collapsible ratio of waste glass powder cement mortar in group F10 fluctuates greatly with the increase in mixing amount. F10C5 has the highest folding pressure, which is 4.1% higher than that of the reference group. The folding pressures of F10C10, F10C15, and F10C20 were reduced by 6.1%, 5.4%, and 15.2%, respectively, compared with the reference group. The flexural–compressive ratio of the F15 group's waste glass powder cement mortar decreased with the increase in dosage. The folding pressures of F15C5, F15C10, F15C15, and F15C20 were reduced by 1.1%, 2.0%, 8.4%, and 13.9% of those of the reference group, respectively.

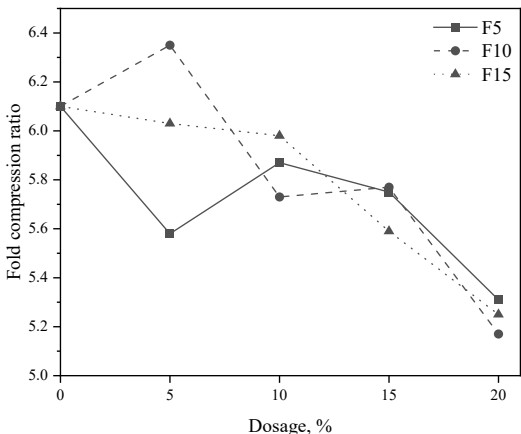

**Figure 17.** The fold compression ratio of waste glass powder cement mortar specimens—28 d.

On the whole, the folding ratio decreases with the increase in waste glass powder content. This indicates that excessive dosage is not conducive to improving the structure of the interfacial transition zone, thus generating C-S-H gel. The primary micro-cracks in the interface area cannot be reduced, so the crack resistance of the specimen cannot be improved.

Under the age of 28 d, the compressive strength of waste glass powder under different grinding times and dosages is shown in Figure 18. Under each grinding time, the compressive strength of 28 d decreased with the increase in dosage. In each dosage, with the increase in grinding time, the compressive strength of 28 d fluctuates. At present, there is no relevant specification for waste glass powder used in cement and concrete. Considering that the main chemical composition of glass is similar to fly ash, it can be according to the relevant fly ash specifications. The determination formula of the strength activity index of fly ash [37] is shown in Equation (1). In this test, the waste glass powder is calculated according to Formula (1): the 28 d compressive strengths of F5C5, F5C10, F5C15, F5C20, F10C5, F10C10, F10C15, F10C20, F15C5, F15C10, F15C15, and F15C20 were 79.9%, 79.4%, 73.6%, 61.9%, 91.3%, 82.4%, 66.7%, 61.4%, 88.6%, 78.7%, 69.6%, and 64.1% of the blank group, respectively. It can be found that the compressive strength of waste glass powder cement mortar with a grinding time of 5–15 min and dosage of 5–10% reached more than 70% of the blank group; therefore, it is feasible for waste glass powder to replace part of cement to prepare a mortar specimen.

$$H_{28} = \frac{R}{R_0} \times 100\% \tag{1}$$

In the formula: $H_{28}$—strength activity index, %.
$R$—test the 28 d compressive strength of mortar, MPa.
$R_0$—compare the 28 d compressive strength of mortar, MPa.

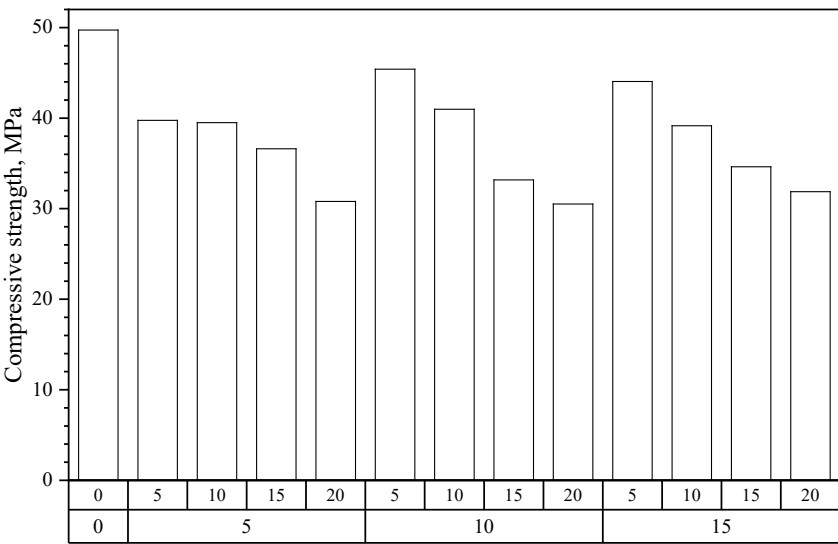

**Figure 18.** Twenty-eight-day compressive strength of waste glass powder under grinding time and amount.

### 3.5. SEM Morphology Analysis

Figure 19 shows the 3 d and 28 d SEM images of the F15C5 waste glass powder mortar specimen. As can be seen from Figure 19a, there are more glass powder particles in the waste glass powder mortar specimen with the age of 3 d. The angular features of waste glass particles indicate that the volcanic ash reaction is not occurring at that time. As can be seen from Figure 19b, the waste glass powder mortar specimen at the age of 28 d shows obvious hydration of the waste glass powder particles, which are completely covered by

hydration products around them, closely bonded with C-S-H cement gel, and the interfacial transition zone is improved. This indicates that the waste glass powder fully has volcanic activity, and the reaction products increase with the increase in age, which reflects that the mortar specimens mixed with waste glass powder have great potential for an increase in compressive strength in the later period.

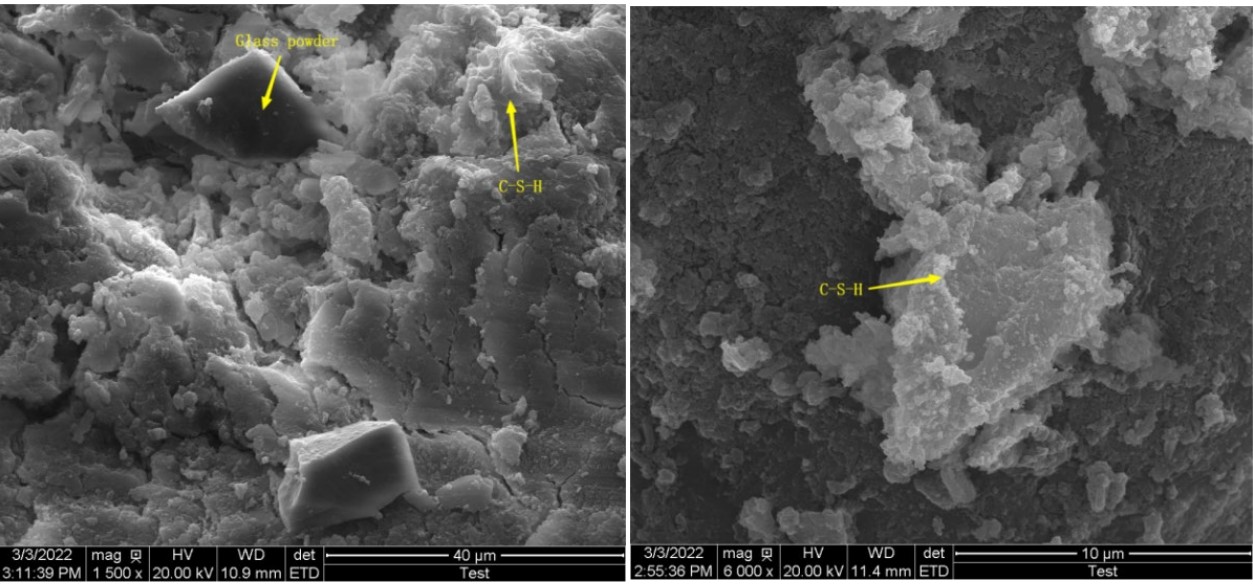

**Figure 19.** SEM image of waste glass powder mortar specimens.

## 4. Conclusions

Through the strength test of the waste glass powder mortar specimens with grinding times of 5–15 min and content of 5–20%, the following conclusions are obtained:

(1) The consistency of cement mortar with waste glass powder increases with the increase in waste glass powder content.

(2) The grinding time has a great influence on the strength of cement mortar. The longer the grinding time is (15 min in this test) and the better the particle size is (average particle size is 13.2 μm), the higher the activity of the waste glass powder is, and the better the compressive and flexural strength is. The F15 group of waste glass powder cement mortar has the best compressive and flexural strength overall.

(3) With the increase in waste glass powder content, the strength of mortar specimens decreased gradually. The strength-changing trend of mortar specimens mixed with waste glass powder is better than that of the blank group. However, the amount of waste glass powder is different. The C5 group of waste glass powder cement mortar has the slightest compressive and flexural strength loss at each age. The C10 group has the best compressive and flexural strength development of cement mortar in the later age. The C15 and C20 groups have no apparent advantages in improving cement mortar's compressive and flexural strength.

(4) The strength development of waste glass powder cement mortar in the early stage is not as good as that in the late stage, which is due to the increase in reaction products in the late stage, and its strength is improved. The folding ratio decreases with the increase in waste glass powder content.

(5) When the grinding time is 5–15 min and the content of waste glass powder is 5–10%, the mortar specimens prepared by replacing part of the cement have the compressive strength activity index of more than 70% of the pure cement mortar specimens in 28 d.

**Author Contributions:** Conceptualization, S.L. and L.S.; investigation, Y.S.; resources, S.L.; data curation, W.Z.; writing—original draft preparation, W.Z.; writing—review and editing, S.L. and L.S.; visualization, J.X. and T.Z.; project administration, S.L.; funding acquisition, S.L. All authors have read and agreed to the published version of the manuscript.

**Funding:** This study is supported by the Changzhou Sci&Tech Program (Grant No. CE20205050 and CZ20210032) and the Natural science research project of Jiangsu colleges and universities (Grant No. 22KJA560005).

**Institutional Review Board Statement:** Not applicable.

**Informed Consent Statement:** Not applicable.

**Data Availability Statement:** Data will be available upon request from the corresponding author.

**Conflicts of Interest:** The authors declare no conflict of interest.

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
