# Peer review of "Studying the Effects of Varied Dosages and Grinding Times on the Mechanical Properties of Mortar"

_sustainability, doi:10.3390/su15075936_

Round 1

Reviewer 1 Report

Comments to the Author

I have read the paper "Study on Mechanical Properties of Waste Glass Powder Cement Mortar". The studies are really very interesting and produced good results, about the study on mechanical properties of waste glass powder cement mortar which is best suited for the “Sustainability Journal”. The manuscript is structured nicely but require some refinement to make it more convenient for the reader to follow the ideas of the authors. However, I would propose that the author make the modest changes listed below.

-        I would suggest to make spelling check throughout the paper and correct it.

-        Also, to modify some sentences that are not correct grammatically and also not clear. However, the overall language of the paper is very simple and clear.

Abstract

Page 1, line 17, kindly write the unit in proper form like in m2, 2 in superscript (670 m2/kg). The abstract clearly defines the objective of the study.

1.     Introduction

This part is well written, however, the description is very short, if possible, kindly elaborate it and add some latest references.

-Page 1, line 26, first sentence "Glass products are widely used in production and life because of their simple production process, light transmission, wear resistance, and low cost." However, I didn’t understand the point about "widely used in production and life" kindly clarify.

-Page 1, line 38, 41, Kindly leave space before and after the reference number in the text. Kindly read the paper through and correct it.

2.     Raw materials and test methods

Test methods and the description are provided nicely.

-Page 2, line 47-48, I suggest to re-write the sentence properly.

Page 2, line 51-53, I suggest to re-write the sentence properly.

Page 2, line 69, Kindly give the proper Unit abbreviation like in 48 r/min, what is “r/min”?

3.     Test Results and Discussion

This part is well discussed considering the test results.

Page 3, Table 2, Please correct the Table 2 bold heading, spelling errors, and write in a consistent manner, some headings written with the first letter in capital letters and others with small letters.

Page 4, Figure 3, in the y and x axis labels, mentioned "Bulk density/%" and Particle size "classification/μm" respectively. What represents \% and \μm? Usually, the unit should be put after the comma, not after the oblique sign. Also, I didn’t understand, in the case of Bulk density, whether you are representing the percentage of different particles with different minutes in % or what.

Page 4, Table 3, Kindly write the heading uniformly, either by writing the first letter in capital letters for all different terms or by writing all headings in small letters. Also, the unit should be written properly, somewhere a bracket is missing, some places have no bracket, and there is no uniformity.

Page 4, line 112, "( sinking amount )" remove the spaces in the bracket.

Page 4, line 118, "figure 4", somewhere in the text, "F" is written in capital letters somewhere in small letters, so kindly make the writing uniform.

Page 5, Figure 4, x and y axis labels, please write the unit with a comma (,) not with an oblique sign (/), like "Grinding time, min", It is more common and clear. Kindly check through the paper in all figures and correct it.

Page10, line 252, Correct FIG. 10, write properly as mentioned in other places in the text.

Page 12, Correct FIG. 13, write properly as mentioned in other places in the text.

After every digit, kindly give the spacing like 5 %, 225 g, 40 mm etc. Kindly check the paper throughout and correct it.

4. Conclusions

Page 13, line 327-328, kindly correct the sentence.

Conclusions are supported in line with the presented results.

References

The latest reference (one) is from 2022, I would suggest to add some latest references if possible.

The format for reference numbers 4, 5, 11, 12,13,14, 16 is not in line with other references. Kindly correct it.

Author Response

Report

Reviewer 1

17.03.2023

I have read the paper "Study on Mechanical Properties of Waste Glass Powder Cement Mortar". The studies are really very interesting and produced good results, about the study on mechanical properties of waste glass powder cement mortar which is best suited for the “Sustainability Journal”. The manuscript is structured nicely but require some refinement to make it more convenient for the reader to follow the ideas of the authors. However, I would propose that the author make the modest changes listed below.

- I would suggest to make spelling check throughout the paper and correct it.

- Also, to modify some sentences that are not correct grammatically and also not clear. However, the overall language of the paper is very simple and clear.

Response: We sincerely thank Reviewer #1 for the time and effort in reviewing our manuscript. We really appreciate all the comments and suggestions, as well as the compliments. In our revised manuscript, we carefully took all the comments and suggestions into account. Below is our point-by-point response.

Abstract

Page 1, line 17, kindly write the unit in proper form like in m2, 2 in superscript (670 m2/kg). The abstract clearly defines the objective of the study.

Response: Thank you for the comment.

On page 1, line 21, we have revised and filled in the correct unit form.

  1. Introduction

This part is well written, however, the description is very short, if possible, kindly elaborate it and add some latest references.

(1) Page 1, line 26, first sentence "Glass products are widely used in production and life because of their simple production process, light transmission, wear resistance, and low cost." However, I didn’t understand the point about "widely used in production and life" kindly clarify.

(2) Page 1, line 38, 41, Kindly leave space before and after the reference number in the text. Kindly read the paper through and correct it.

Response: Thank you for the comments and suggestions. In this part, we have described in detail and added some latest references.

(1) Page 1, lines 29-30, glass can be used as glassware, chemical testing instruments, and other production applications and as glass, glass jewelry, and other applications in life.

(2) We have left blanks before and after the reference number of the text.

  1. Raw materials and test methods

Test methods and the description are provided nicely.

(1) Page 2, line 47-48, I suggest to re-write the sentence properly.

(2) Page 2, line 51-53, I suggest to re-write the sentence properly.

(3) Page 2, line 69, Kindly give the proper Unit abbreviation like in 48 r/min, what is “r/min”?

Response: Thank you for the comment and suggestions.

(1) Page 2, lines 66-69, The writing has been polished and revised during revision.

(2) Page 2, lines 76-78, The writing has been polished and revised during revision.

(3) Page 3, line 102, The writing has been revised during revision.

  1. Test Results and Discussion

This part is well discussed considering the test results.

(1) Page 3, Table 2, Please correct the Table 2 bold heading, spelling errors, and write in a consistent manner, some headings written with the first letter in capital letters and others with small letters.

(2) Page 4, Figure 3, in the y and x axis labels, mentioned "Bulk density/%" and Particle size "classification/μm" respectively. What represents \% and \μm? Usually, the unit should be put after the comma, not after the oblique sign. Also, I didn’t understand, in the case of Bulk density, whether you are representing the percentage of different particles with different minutes in % or what.

(3) Page 4, Table 3, Kindly write the heading uniformly, either by writing the first letter in capital letters for all different terms or by writing all headings in small letters. Also, the unit should be written properly, somewhere a bracket is missing, some places have no bracket, and there is no uniformity.

(4) Page 4, line 112, "( sinking amount )" remove the spaces in the bracket.

(5) Page 4, line 118, "figure 4", somewhere in the text, "F" is written in capital letters somewhere in small letters, so kindly make the writing uniform.

(6) Page 5, Figure 4, x and y axis labels, please write the unit with a comma (,) not with an oblique sign (/), like "Grinding time, min", It is more common and clear. Kindly check through the paper in all figures and correct it.

(7) Page10, line 252, Correct FIG. 10, write properly as mentioned in other places in the text.

(8) Page 12, Correct FIG. 13, write properly as mentioned in other places in the text.

(9) After every digit, kindly give the spacing like 5 %, 225 g, 40 mm etc. Kindly check the paper throughout and correct it.

Response: Thank you for the comment and suggestion.

(1) Page 4, table 2, The writing has been revised during revision.

(2) Page 5, figure 3, Place the unit after the comma in the modification. The unit of volume density is %, which means the percentage of volume density corresponding to different particle size classifications.

(3) Page 5, table 3, Capitalizes the first letter of all different terms. The unit is written right, and the place without brackets is re-modified.

(4) Page 5, line 147, "( sinking amount )" have removed the spaces in the bracket.

(5) Page 6, line 153, Unified writing into ' Figure ' during the paper.

(6) Figure 3-18, x and y axis marked, units have changed to a comma.

(7) Page 17, Figure 16-17, The writing has been revised during revision.

(8) Page 19, Figure 19, The writing has been revised during revision.

(9) We have checked the paper and given the spacing after every digit.

  1. Conclusions

Page 13, line 327-328, kindly correct the sentence.

Conclusions are supported in line with the presented results.

Response: Thank you for the comment and suggestion.

Because of typesetting, we did not find the corresponding sentence to be modified on page 13, lines 327-328.

References

The latest reference (one) is from 2022, I would suggest to add some latest references if possible.

The format for reference numbers 4, 5, 11, 12,13,14, 16 is not in line with other references. Kindly correct it.

Response: Thank you for the comment and suggestion.

We have added some latest references during the revision. Reference number 4 is duplicated with reference number 8, and reference number 4 has been deleted. The format of reference number 5 has been modified; see reference number 12. The format of reference numbers 12, 13, 14, and 15 has been modified; see reference numbers 20, 21, 22, and 23. The format of reference number 16 has been modified; see reference number 30.

Reviewer 2 Report

This paper investigated the effects of using varied dosages of grinded waste glass powder on the physical properties of cement mortar. In general, the study is useful and well-presented the case of study. However, there are some comments that need to be considered to further improve the manuscript:

1- The title is too general which needs to highlight the main novelty of the current study, which is studying the effects of varied dosages and grinding time on the mechanical/physical properties of the mortar.

2. It needs to clarify clearly, why this study is suggested in the first place. What is the problem that needs to be solved?.......etc.

3. The background and the main problem of the statement related to this study need to be highlighted and discussed in the abstract.

4. The Introduction section does not motivate the current research problem very well, thus it must be improved by technically discussing additional related studies.

5. In general, the background of this study is well presented, however, the existing reference is quite old, and more recent studies are needed to be presented and discussed, specifically those released in the latest 3 years (2020 up to date).

6. Table 2: double check the value of the average particle size of grinding time 10min.

7. Section 3.2 presented the Effect of grinding time of WGP with 5% replacement in Figures 5 and 6, what about the other replacement dosages (10%, 15%, and 20%)??, also needs to be discussed.

8. Sections 3.2 and 3.3 need to update the end of their title to read’ compressive and flexural strength of mortar”.

9. Figures 5 to 8 need to mention the related dosage value and/or grinding time in their title. Example: “Figure 8. Effect of different dosages on flexural strength of mortar- F15 group”.

10. The introduction and/or conclusions can be more attractive by highlighting the limits of the current study.

Round 2

Reviewer 2 Report

The authors addressed the given comments adequately, and the revised manuscript accepted